# 3D-Printed Chitosan-Based Scaffolds with *Scutellariae baicalensis* Extract for Dental Applications

**DOI:** 10.3390/pharmaceutics16030359

**Published:** 2024-03-04

**Authors:** Magdalena Paczkowska-Walendowska, Ioanna Koumentakou, Maria Lazaridou, Dimitrios Bikiaris, Andrzej Miklaszewski, Tomasz Plech, Judyta Cielecka-Piontek

**Affiliations:** 1Department of Pharmacognosy and Biomaterials, Poznan University of Medical Sciences, 60-806 Poznan, Poland; jpiontek@ump.edu.pl; 2Laboratory of Polymer Chemistry and Technology, Department of Chemistry, Aristotle University of Thessaloniki, 541 24 Thessaloniki, Greece; iwanna.koumentakou@gmail.com (I.K.); marlazach@chem.auth.gr (M.L.); dbic@chem.auth.gr (D.B.); 3Faculty of Mechanical Engineering and Management, Institute of Materials Science and Engineering, Poznan University of Technology, 61-138 Poznan, Poland; andrzej.miklaszewski@put.poznan.pl; 4Department of Pharmacology, Medical University of Lublin, 20-080 Lublin, Poland; tomasz.plech@umlub.pl; 5Department of Pharmacology and Phytochemistry, Institute of Natural Fibres and Medicinal Plants, 60-630 Poznan, Poland

**Keywords:** 3D printing, chitosan, gelatin, *Scutellariae baicalensis* extract

## Abstract

The plant material *Scutellariae baicalensis radix*, which is rich in flavones (baicalin), possesses antibacterial, antifungal, antioxidant, and anti-inflammatory properties. This work aimed to develop a 3D-printed chitosan-based hydrogel rich in *Scutellariae baicalensis* extract as an innovative approach for the personalized treatment of periodontal diseases. Chitosan-based hydrogels were prepared, and the printability of the prepared hydrogels was determined. The hydrogel with 2.5% *w*/*v* of high molecular-weight chitosan (CS), 2% *w*/*v* gelatin (Gel), and 10% *w*/*w* of extract (Ex) presented the best printability, producing smooth and uniform scaffolds. It was proved that the CS/Gel/Ex hydrogel was stabilized by hydrogen bonds and remained in amorphous dispersion in the 3D-printed structures (confirmed by ATR-FTIR and XRPD). Due to the amorphization of the active substance, a significant increase in the release of baicalin in vitro was observed. It was demonstrated that there was an initial burst release and a continuous release profile (*n* = 3). Higuchi kinetic was the most likely baicalin release kinetic. The second fit, the Korsmeyer–Peppas kinetics model, showed coupled diffusion of the active ingredient in the hydrated matrix and polymer relaxation regulated release, with n values ranging from 0.45 to 0.89. The anti-inflammatory properties of 3D-printed scaffolds were assessed as the ability to inhibit the activity of the hyaluronidase enzyme. Activity was assessed as IC_50_ = 63.57 ± 4.98 mg hydrogel/mL (*n* = 6). Cytotoxicity tests demonstrated the biocompatibility of the material. After 24 h of exposure to the 2.5CS/2Gel/10Ex scaffold, fibroblasts migrated toward the scratch, closed the “wound” by 97.1%, and significantly accelerated the wound healing process. The results render the 3D-printed CS/Gel/extract scaffolds as potential candidates for treating periodontal diseases.

## 1. Introduction

Periodontal disease represents a significant global burden on oral health, with estimated prevalence ranging from 20% to 50% worldwide [1]. The host’s immune system’s reaction to the aggressiveness of periodontopathogen bacteria is a major contributing component in the multifactorial etiology of periodontitis, an inflammatory disease of the periodontal tissues. This condition is characterized by the gradual loss of alveolar bone and the progressive loss of periodontal attachment, both of which might eventually result in tooth loss [2]. The mainstay of initial non-surgical periodontal therapy is scaling and root planing, together with a review of home care. It is possible to use either traditional resective or modern regenerative surgery for residual locations with active periodontitis at periodontal re-evaluation [3]. Due to limited treatment methods, looking for innovative therapy proposals is necessary.

Since scaffolds are considered essential for effective tissue creation, tissue engineering holds the answer to both bone regeneration and periodontal tissue regeneration [4]. Consequently, the implantation of endosseous implants and scaffolds has transformed contemporary dentistry, as more and more patients seek periodontal therapy using this approach. The scaffolds must have architecture and motifs that can promote the cell attachment and the growth of local stem cells and or/osteoprogenitor cells [5].

The creation of three-dimensional (3D) scaffolds involves the application of many processes (traditional approaches such as gas foaming, particle leaching, freeze drying, phase separation, fiber meshes/fiber bonding, melt molding, and solution casting) with the 3D-printing method to be the most effective technique [6]. The ability to create scaffolds with clearly defined structures and physical orientations that promote homogenous cell growth rate and enable the restoration of homogeneous, well-formed tissue is the primary benefit of the 3D-printing process [7,8]. Moreover, the used biomaterials for the manufacturing of the scaffolds must be specific for the application as well as be in balance with the surrounding environment to guarantee that the damaged area is replaced with a functional, healthy tissue that is similar to the original one and does not produce healing scars [9]. The use of such a dressing is possible at every stage of periodontal treatment, from surface application to treat local inflammation, to placement directly in the gingival pocket, where it can have antimicrobial effects to remove biofilm, to the previously mentioned inflammation-relieving effects, and finally to regeneration of bone defects [5]. Although 3D-printing technology contributes significantly to regenerative dentistry and to periodontal treatment, it presents several disadvantages. Some 3D-printed scaffolds are brittle with low mechanical stability, and as a result, the damaged tissue cannot be fully regenerated. In other cases, the biomaterials used are unable to fabricate scaffolds with the desired architectural structure, hindering their ability to provide a proper template for effective cell interaction [10]. Furthermore, other research groups have manufactured 3D-printed scaffolds with highly mechanical properties and integrity but a low biocompatibility [9].

Chitosan (CS) is widely used in the treatment of periodontal diseases as well as for the fabrication of 3D-printed scaffolds for tissue engineering applications [11,12,13]. CS is a cationic polymer derived from the partial deacetylation of chitin and possesses exceptional characteristics, including biocompatibility, biodegradability, anti-inflammatory activity, and antimicrobial properties [14]. Furthermore, CS promotes hemostasis due to direct electrostatic interactions between negatively charged red blood cells and platelets and positively charged chitosan molecules, leading to blood coagulation [15]. As a natural polymer, gelatin (Gel) is also known for its suitability for hydrogel printing due to its easy preparation and manipulation [16]. Gel presents an almost identical composition to collagen, the main component of the native extracellular matrix (ECM). It contains the Arg–Gly–Asp (RGD)-like sequence that promotes cell adhesion and migration [17] and enhances wound healing rate [18,19]. It has been proved that when Gel and CS are blended, the structure formed can affect the polycationic CS interaction with the anionic cell surface. Cell adhesion, cellular bioactivity, the tissue remodeling process, and ultimately the quality of the regenerated tissue are all impacted by the combination of CS and Gel [16,20].

Additionally, incorporating an active plant extract into the structure of a 3D-printed scaffold could increase the effectiveness of the promoted method of treating periodontal diseases.

Recently, in the treatment of periodontitis, more and more attention is paid to the use of plant materials due to their multidirectional action profile [21]. One such plant material is *Scutellariae baicalensis* radix (Baikal skullcap root), rich in baicalin, baicalein, and wogonin. In the oral cavity, *S. baicalensis* radix modifies the inflammatory processes by suppressing the expression of proinflammatory mediators such IL-1β, IL-6, IL-8, and TNFα in gingival tissues, reducing alveolar bone degradation and promoting the healing of periodontal structures [22]. Additionally, *S. baicalensis* exhibits strong antibacterial activity against oral pathogens such as *Streptococcus mutans*, *S. Fusobacterium nucleatum*, *Aggregatibacter actinomycetemcomitans*, and *Porphyromonas gingivalis* [23].

Our earlier research’s findings verified that *S. baicalensis radix* extract and CS worked in concert to promote antioxidant activity and inhibit hyaluronidase [22]. Therefore, this research aims to combine all properties of these ingredients (CS, Gel, and *S. baicalensis* radix extract) and to manufacture a hydrogel that can be used for the fabrication of 3D-printed scaffolds tailored to the individual state of each patient, allowing the development of personalized dressing with proven anti-periodontal effect.

## 2. Materials and Methods

### 2.1. Plant Materials

Plant raw material, *Scutellariae baicalensis radix*, was purchased from NANGA (Zlotow, Poland); the country of origin: China (Lot No. 243042021).

### 2.2. Chemicals

Baicalin (Phyproof^®^ Reference Substance) was obtained from Sigma-Aldrich (Poznan, Poland). Chitosan in powder (high molecular weight = HMW; with viscosity 800–2000 cps of 1% solution in 1% *v*/*v* acetic acid; degree of deacetylation ≥ 75%) was supplied by Sigma-Aldrich (Poznan, Poland), while gelatin (Ph.Eur.) was supplied by Polgen (Warsaw, Poland). Sigma-Aldrich (Poznan, Poland) provided the reagents for the artificial saliva solution for the dissolution studies (potassium chloride, sodium chloride, di-potassium hydrogen orthophosphate, magnesium chloride, calcium chloride, and xylitol) and the activity assays (sodium chloride, bovine serum, hexadecyltrimethylammonium bromide (CTAB), and hyaluronic acid (HA)). Water and HPLC grade acetonitrile were purchased from Merck in Darmstadt, Germany. Using a Direct-Q 3 UV Merck Millipore purification system, high-quality and ultra-high-quality pure water were produced.

### 2.3. Preparation Scalcup Root Extract

A concentration of 80% ethanolic extract by ultrasound-assisted extraction was prepared according to procedure described previously, with a verified content of active compounds: baicalin 2610.24 ± 0.68, baicalein 323.40 ± 0.14, and wogonin 43.10 ± 0.01 µg per 100 mg of lyophilized extract [22].

### 2.4. Preliminary Studies of 3D-Printed Scaffolds Preparation

First, hydrogels were prepared according to the composition shown in Table 1. First, Gel was dissolved (temperature 60 °C for 30 min). Then, the extract was added into a gelatin aqueous solution under continuous magnetic stirring (30 °C for 15 min), and subsequently, chitosan powder was added before the dropwise addition of acetic acid (1% *v*/*v*) until complete homogenization. The obtained hydrogels were placed into a sterile printing cartridge and were centrifuged for 15 min at 4000 rpm to remove any air bubbles (NÜVE NF 800, NÜVE, Ankara, Turkey).

The mesh scaffold model’s STL file was utilized for 3D printing, and Slic3r software 1.3.0 (https://slic3r.org, accessed on 27 March 2023) was employed to slice the STL sample. The scaffolds were created as cubic mesh scaffolds (20 mm length × 20 mm width) with 10 layers and an angle of 0° for the printability investigation. The prepared hydrogels were printed using a pneumatic extrusion-based 3D Bioprinter (INKREDIBLE+ 3D Bioprinter, Cellink; Gothenburg, Sweden) with plastic nozzles with inner diameters of 0.41 mm (G22), while a minimum pressure of 44, 85, or 135 kPa for P1, P2, and P3 correspondingly, as well as a printing speed of 25 mm/s and a printing temperature of 25 °C were needed to extrude a continuous strut.

The viscosity values of hydrogels were evaluated at shear rate from 5 to 20 1/s using a LV-3 spindle Brookfield DV2T viscometer at 25 °C, assessed using Rheocalc T software. All the experiments were performed in triplicate.

Two printing parameters were investigated to ascertain the printability of each hydrogel. To compute the uniformity factor and porosity factor, three distinct areas and fibers were examined using a microscope (ZEISS SteREO Discovery.V20 Motorized Stereo Microscope, Jena, Germany) and ImageJ (https://imagej.net/nih-image/, accessed on 27 March 2023) [24].

The uniformity of the printed scaffolds in comparison to the planned STL file was assessed using the uniformity factor (*U*).
$$U=\frac{diameter\;of\;printed\;sample}{diameter\;of\;theoretical\;sample}$$

To ascertain whether the printed pores matched the intended square pores, the pore factor (*Pr*) was employed.
$$Pr=\frac{pore\;area\;of\;printed\;sample}{pore\;area\;of\;theoretical\;sample}$$

Based on this research, the best 3D prints were selected for further research.

### 2.5. Characterization of the 3D-Printed Scaffolds

#### 2.5.1. Microscopic Analysis

The microphotograph was taken by the VHX 7000 Keyence microscope (Mechelen, Belgium) using the digital depth composition mode at 16-bit high dynamic range and color height profiling.

#### 2.5.2. Scanning Electron Microscopy (SEM)

SEM was used to examine the scaffold’s surface morphology. Samples were carbon-coated before examination; then, samples were checked using a TESCAN MIRA’s 3th generation Scanning Electron Microscope (SEM) with FEG Schottky electron emission source (Brno-Kohoutovicem, Czech Republic).

#### 2.5.3. Fourier Transform Infrared Spectroscopy with Attenuated Total Reflectance (ATR-FTIR)

Using an IRTracer-100 (Shimadzu, Kyoto, Japan) spectrophotometer and LabSolutions IR software (version 1.86 SP2, Shimadzu, Kyoto, Japan), the ATR-FTIR spectra were obtained in absorbance mode, covering a range of 400 to 4000 cm^−1^. The spectrometer was configured with 400 scans, Happ-Genzel apodization, and a resolution of 4 cm^−1^.

#### 2.5.4. X-ray Diffraction (XPD)

A Panalytical Em-pyrean (Almelo, the Netherlands) X-ray diffraction (XRD) apparatus with a copper anode (CuKα—1.54 Å) in a Brag-Brentano reflection mode setup at 45 kV and 40 mA settings was used to study the samples’ structure. The measurement parameters were adjusted consistently between 3 and 60°, with a 45 s step between each degree.

#### 2.5.5. Swelling Capacity

The produced scaffolds’ ability to swell was assessed by calculating the artificial saliva solution’s water sorption aptitude (pH = 6.8) [22]. Each dry scaffold was carefully weighed (*W_d_*) and placed in a solvent-filled baker. The samples were placed on filter paper to remove extra surface water at predefined intervals of time (15 min, 30 min, 1 h, 2 h, 4 h, 6 h, and 24 h), and they were then weighted once more (*W_f_*). All the experiments were performed in triplicate. The swelling ratio was calculated according to the equation:$$Swelling\;index\;\left(\%\right)=\frac{\left(W_{f}-W_{d}\right)}{W_{d}}\times 100$$

#### 2.5.6. Release of Active Components

Three-dimensional-printed scaffolds were subjected to dissolve experiments using the Agilent 708-DS dissolving equipment (Agilent Technologies; Santa Clara, CA, USA). A typical basket method was used, with 50 rpm and stirring at 37 ± 0.5 °C. Samples were added to 300 milliliters of the artificial saliva solution (pH 6.8), which mimics the pH of the oral cavity. Liquid samples were taken at prearranged intervals, and the same volume of temperature-stabilized media was replaced. The samples were filtered using a 0.45 µm mesh nylon membrane filter. All the experiments were performed in triplicate. Using the previously developed HPLC procedure [22], the amounts of baicalin in the filtrated acceptor solutions were ascertained. The Kinetex^®^ C18 column (Phenomenex, Poland) with a 5 μm particle size and dimensions of 100 mm by 2.1 mm was used for the separations. A diode array detector operating at 280 nm wavelength maxima (*λ*_max_) was used for the detection. Phosphoric acid 0.1% (A) and acetonitrile (B) were combined to create the mobile phase, which was eluted in two stages: 10% B for 20–22 min and 10% B for 0–20 min. The temperature of the column was kept at 30 °C, and the mobile phase flow rate was 1.0 mL/min.

The first-order, zero-order, Higuchi, and Korsmeyer–Peppas models were fitted to the obtained active compound release patterns [25].

#### 2.5.7. Anti-Inflammatory Activity

The hyaluronidase inhibition properties of the 3D-printed scaffold, with the previously disclosed method [22], was determined using a turbidimetric methodology. In brief, after mixing 25 µL of acetate buffer (50 mM, pH 7.0, with 77 mM NaCl and 1 mg/mL of albumin), 25 µL of enzyme (30 U/mL of acetate buffer, pH 7.0), 15 µL of acetate buffer (pH 4.5), and 10 µL of extracts, the mixture was incubated for 10 min at 37 °C. Acetate buffer (pH 4.5) containing 25 µL of HA was added, and the mixture was incubated for 45 min at 37 °C. Further, 200 µL of 2.5% CTAB in 2% NaOH (pH 12) was added to precipitate the undigested HA. The mixture was allowed to stay at room temperature for ten minutes. The test was performed in 6 repetitions. The absorbance at λ = 600 nm was used to calculate the inhibition percentage, using the following formula:$$\%\;inhibition\;activity=\frac{T_{S}-T_{C}}{T_{H}-T_{C}}\times 100\%$$
where *T_S_*—absorbance of the enzyme + HA + extract, *T_C_*—absorbance of the enzyme + HA, and *T_H_*—absorbance of the HA + extract.

#### 2.5.8. Microbiological Activity

For Gram-positive bacteria (*Streptococcus mutans*), the minimum inhibitory concentration (MIC) was assessed. MIC was determined by the micro-dilution method using 96-well plates (Nest Scientific Biotechnology, Woodbridge, NJ, USA). The process has been outlined in earlier articles [26].

#### 2.5.9. Cytotoxicity Assay

The viability of human normal skin fibroblasts (Hs27 cells) incubated for 24 h with hydrogel base (2.5CS/2Gel), hydrogel-based scaffold (2.5CS/2Gel/10Ex), *Scutellaria baicalensis* extract (Extract), and baicalin (Bai) was examined using the MTT assay. Samples of 2.5CS/2Gel and 2.5CS/2Gel/10Ex were mixed with the culture medium to obtain 1% solution (0.1 mL of 2.5CS/2Gel or 2.5CS/2Gel/10Ex + 9.9 mL of medium). Concentration of the Extract was adjusted to its content in 2.5CS/2Gel/10Ex, while Bai was tested at its concentration determined in the *Scutellaria baicalensis* extract. Human normal skin fibroblasts (Hs27) were cultured as described in 2.5.9. subchapter. Next, Hs27 cells were seeded into 96-well sterile plates at a cell density of 1 × 10^5^ cells/mL. After 24 h of incubation, the medium was removed from each well and then cells were incubated for the next 24 h with the respective samples (in DMEM containing 2% FBS). Control cells were cultured only with the medium containing 2% FBS. Toxicity of the compounds was evaluated using the MTT assay, which is based on the conversion of 3-(4,5-dimethylthiazol-2-yl)-2,5-diphenyltetrazolium bromide (MTT) into dark-blue formazan crystals. Briefly, after 24 h of incubation of cells with the tested samples, the medium was removed from the plates. Cells were washed with PBS, and then, 100 µL of medium containing 10% MTT solution (5 mg/mL) was added to each well. Next, Hs27 cells were incubated for 4 h at 37 °C in an atmosphere of 5% CO_2_, and, subsequently, 100 µL (per well) of 10% SDS buffer solution was added to dissolve the formazan crystals. After an overnight incubation, the absorbance was measured at 570 nm using a microplate reader (Epoch, BioTek Instruments, Winooski, VT, USA). Experiments were repeated twice, and the measurements in each experiment were run in triplicate.

#### 2.5.10. Wound Healing Properties

Using the scratch assay, the ability of hydrogel and its components to heal wounds was investigated on Hs27 cells. Human skin fibroblasts Hs27 (CRL-1634) were obtained from the American Type Culture Collection (Manassas, VA, USA). The cells were cultured in DMEM-high glucose supplemented with 10% FBS, penicillin (100 U/mL), and streptomycin (100 µg/mL). Prior to the experiment, the Hs27 cells were planted at a density of 1 × 10^5^ cells/mL on 6-well culture plates (Corning Inc., Corning, NY, USA) after being separated using trypsin/EDTA. After the cell confluence approached 90%, a vertical linear scratch was formed in the monolayer with a sterile pipette tip. To eliminate any leftover cell debris, three rounds of washing with phosphate-buffered saline (PBS) were followed by the addition of a new medium containing hydrogel base (2.5CS/2Gel), hydrogel-based scaffold (2.5CS/2Gel/10Ex), *Scutellaria baicalensis* extract (Extract), baicalin (Bai), or 2% FBS only (control group) to the corresponding wells. After that, images of the scratch were taken at 0 h and 24 h using an Olympus CKX53 microscope equipped with an XM10 digital camera (Olympus). The experiments were run in triplicate. The open wound area was measured using NIH ImageJ software (Bethesda, Rockville, MD, USA) (https://imagej.net/nih-image/, accessed on 20 February 2024) and considered as 100% at the beginning of the experiment (at 0 h). The wound closure % was calculated using the following formula:$$Wound\;closure\left(\%\right)=\frac{open\;wound\;area\;at\;0\;h\;-\;open\;wound\;area\;at\;24\;h}{open\;wound\;area\;at\;0\;h}\times 100\%$$

### 2.6. Statistical Analysis

The statistical analysis was carried out using Statistica 13.3 (StatSoft Polska Sp. z o.o.; Krakow, Poland). The Shapiro-Wilk test was run to see if the data were normal. The ANOVA and Tukey’s and/or Duncan’s post hoc range tests for multiple comparisons were used to examine the variances between the mean values. At *p* < 0.05, differences across groups were deemed significant.

## 3. Results and Discussion

The main challenge in the preliminary study was selecting the appropriate combination of materials in which the extract would be encapsulated and present high 3D printability. In the presented work, the influence of different concentrations of chitosan on the properties of the prepared hydrogel and its printing possibilities was assessed, while the concentration of the remaining ingredients remained at the same level [24]. Previous findings by Zhang et al. show a significant dependence of the CS/Gel ratio, where the best osteoblastic responses, including cell adherence progress, spreading area, proliferation, and focal adhesion rate-related gene expression, were demonstrated when CS is in a larger amount than Gel [27]. Therefore, the assumptions of the composition are part of this action, which is important from the point of view of complex odontogenic infections.

The flow capacity of bioinks is a crucial factor that affects the printability of inks. Shear thinning behavior is generally what an ideal ink for 3D printing should exhibit [28]. For consistent strands and to prevent nozzle clogging, the materials used should have just the right amount of viscosity to hold their shape during the printing process. The high viscosity of the ink helps both during and after 3D printing because the prepared filament is stiff and challenging to flow and spread. As a result, the final sample and the filament maintain their original shapes [29]. Inks with viscosity values greater than 10,000 Pa·s are considered excessively viscous, while those with viscosity values less than 100 Pa·s are considered excessively fluidic [30]. Since CS naturally creates high viscosity, non-Newtonian gels, which are essential for a successful 3D-printing application, have been used in numerous 3D-printing applications [12], also in the presented studies.

Therefore, the three newly formed formulations were initially tested for their viscosity through rheological measurements via increasing the shear rate (Figure 1). It is observed that the increase in the CS concentration in the hydrogels leads to an increase in the viscosity values (Figure 1). This happens due to the increase in the intensity of electrostatic interactions of the positively charged amino groups as the CS macromolecular chain is highly stretched in an acidic environment [31]. Based on the above values, formulations 2CS/2Gel/10Ex and 2.5CS/2Gel/10Ex meet the viscosity requirements [30].

Examining the printability of scaffolds proves to be a valuable approach to assessing the capabilities of hydrogel-based inks for applications in 3D bioprinting. To qualify as printable, an ink must fulfill some requirements, including the ability to be extruded and ensure accurate shape fidelity to the construct.

The viscosity of the printing ingredients controls the air pressure (*P*) needed for extrusion in 3D-bioprinting applications. Any escalation in the viscosity of hydrogels inevitably translates into an elevated requirement for *P* during their extrusion. As presented by the viscosity results (Figure 1), the composition of the hydrogels and specifically the CS content determined the prepared hydrogels’ viscosity, requiring a higher *P* for their extrusion (Table 2). As the hydrogel viscosity increased, a higher minimum extrusion pressure was needed.

It was found that more hydrogel was extruded from the nozzle when the pressure was above the minimum pressure required. On the other hand, the hydrogels could not be extruded when lower pressure than the required pressure was applied.

The G22 nozzle size was used to print a filament at the lowest extrusion pressure at 25 °C. The strand uniformity factor (*U*) for each hydrogel was then calculated (Table 3).

During 3D printing, the resolution was controlled. Each formulation was able to produce squared grid structures with excellent shape fidelity. When measuring uniformity factors, the 2.5CS/2Gel/10Ex hydrogels showed their *U* values closest to the theoretical value of 1 and did not differ statistically from 3CS/2Gel/10Ex. The 2CS/2Gel/10Ex hydrogels spread after extrusion due to their high fluidity and lowest viscosity of all the samples. This caused the filament to bend and the printed material to collapse laterally, yielding the significantly higher *U* values. The 2.5CS/2Gel/10Ex scaffolds had square pores and the *Pr* value was closest to 1. In the case of 2CS/2Gel/10Ex, the material spread after extrusion, and for 3CS/2Gel/10Ex, a discontinuous material flow was observed, which caused the formation of rectangular pores instead of square pores, resulting in higher values of pore factors (Table 3). Based on the results of homogeneity analyses as well as the intact mesh surface indicating high dimensional stability after printing, it was determined that the best hydrogel–ink composition to produce 3D-printed scaffolds is the 2.5CS/2Gel/10Ex (P2) hydrogel.

The best formulation was selected and characterized in terms of microscopic (Figure 2) and structural (Figure 3) analysis.

Based on the microscopic image (Figure 2), it is clearly visible that the shape of the print is uniform, and extrusion of the 2.5CS/2Gel/10Ex (P2) hydrogel produces an equal-sized square-shaped mesh scaffold as described above. At the same time, the uniformity of the thickness of the produced dressing is noticeable. The average thickness of the scaffolds in the interior, excluding the outermost line, was within the range of 620–750 μm (Figure 2b). Furthermore, SEM micrographs (Figure 2c) show the microstructure and morphology of the printed scaffolds. In the case of manufactured samples, their surface is smooth, which confirms the correctness of the process.

Characteristic bands of extraction and CS have been previously described [22]. In summary, the lyophilized extract of *S. baicalensis* exhibited distinctive bands at 3330, 1720, and 1660 cm^−1^, which were associated with the stretching vibration of the O-H, -COOH, and C=O groups. Additionally, signals at 1600 and 1580 cm^−1^ were linked to stretching of the aromatic rings in the flavone structure due to C=C vibration. The wide bands within the 1200–900 cm^−1^ range were linked to the different stretching vibrations of saccharide C-O bonds. Regarding the chitosan spectrum, the peak at 2900 cm^−1^ from C-H vibrations and at 1650 cm^−1^ from N-H vibrations are observed, along with the distinctive bands at 3360 and 3300 cm^−1^, originating from the stretching vibrations of the O-H and N-H groups. The amide B band, which is visible at 3080 cm^−1^, as well as the amide I, II, and III bands, which are visible at the 1646 cm^−1^, 1550 cm^−1^, and 1239 cm^−1^ wavenumbers, respectively, are distinctive groups found in the structure of gelatin. The stretching vibrations of CN and CO are responsible for the amide I band. The N-H group’s deformation vibrations and the C-N bonds’ stretching vibrations are attributed to the amide band II. Moreover, additional bands that indicated the presence of the amide III were associated with the vibrations of bending N-H bonds and stretching C-N bonds [32]. Subsequently, the 2.5CS/2Gel/10Ex scaffolds were examined (Figure 3a). All the scaffolds exhibited the overlapping curve that is thought to be caused by the hydrogen bonds between amino and hydroxyl groups found in the CS and extract structure. No new peak could be seen when compared to the spectra of CS, Gel, and composite scaffolds. This suggests that the composite scaffolds may have had a role in forming hydrogen bonds between and within the CS-Gel molecules, which caused the observed shift in the peaks. No further changes in structure were observed.

The 2.5CS/2Gel/10Ex scaffolds were further analyzed using XRPD measurements (Figure 3b). The behavior of the extract, demonstrating a substantial broadening of diffraction peaks, which at low intensity implies its amorphousness, was the same for Gel, while CS is a semi-crystalline polysaccharide with two characteristic peaks at 10 and 20° [33]. There was a decrease in the intensity of the CS peaks when testing 2.5CS/2Gel/10Ex. This may happen due to the interactions of CS/Gel mixtures caused by electrostatic interactions and hydrogen bonds as previously reported [34]. It seems that the addition of gelatin in chitosan solutions promotes the amorphous structure of the polymers and hinders crystallization of chitosan. The absence of the crystalline peaks of chitosan, as well as the lack of a new peak in the composites, shows that CS, Gel, and extract were adequately compatible; these findings further support the idea that the as-synthesized polymeric matrix system does not contain a new phase [35]. Moreover, the amorphous nature of the diffractograms for scaffolds suggests that there is no tendency for re-crystallization, which is most likely caused by the high moisture content and durability of the CS/Gel matrix [36].

The capacity of polymer materials meant for dental dressings to swell in an aqueous environment is a crucial characteristic, particularly when it comes to exudates. For the 3D-printed scaffold, the swelling index was 39.04 ± 2.51% after 24 h (Figure 4). The swelling process behaves in two stages: a rapid swelling phase that lasts for the first hour, followed by a continuous swelling phase. The obtained value is lower than for neat CS, described in the literature [37], due to adding the extract, which determines a different biological effect. In addition, it has been reported that chitosan appears to have a low swelling index at a pH above its pKa = 5.6 because of its hydrophobic nature [38]. However, a too high swelling index in the case of buccal administration would also not be advisable due to patient discomfort [39]. Therefore, the result obtained is satisfactory; it proves the possibility of swelling and absorbing any exudates and enables the effective release of the extract’s active compounds.

Next, dissolution behavior of baicalin from the scaffold containing the lyophilized extract was examined (Figure 5). Due to the amorphization of baicalin via extract lyophilization, a significant increase in the release of baicalin in vitro was observed (black line for lyophilized extract vs. red line for crystalline baicalin). Concerning the release of the extract from the 3D-printed scaffolds (blue line), it seems that there is an optimization of its in vitro release behavior since the blends contribute to a release in a more controlled way. Furthermore, two phases can be noticed. An initial burst baicalin release during the first two hours is followed by a sustained release over 6 h.

The most likely baicalin release kinetic was the Higuchi kinetic (Table 4). The second fit, the Korsmeyer–Peppas kinetics model, showed that drug release was regulated by a combination of polymer relaxation and drug diffusion in the hydrated matrix, with n values ranging from 0.45 to 0.89. Chitosan is unquestionably suitable as a material for the regulated release of biologically active substances [40].

Uniformity (1.5048 ± 0.0542) and porosity factors (1.3592 ± 0.0523) of post-release dressings were also assessed. The increase in parameter values results from the swelling properties of the dressings, but importantly, the standard deviation of the results remained very low, which proves the same behavior of individual dressings.

CS inhibited the hyaluronidase enzyme’s activity and extract [22], demonstrating the anti-inflammatory activity, which was evaluated before (Table 5). Therefore, the activity on 3D-printed material was assessed and resulted in IC_50_ = 63.57 ± 4.98 mg hydrogel/mL. Reduced activity towards the starting substances may result from combining the ingredients and diluting them in the mixture mass in an acidic environment. However, the results obtained show high anti-inflammatory activity, which guarantees biological effectiveness.

Considering that infections are one of the main causes of dental health problems, using 3D-printed hydrogels with antimicrobial qualities has benefits for treating periodontal abnormalities [41]. The potential of the extract and CS against bacterial and fungal species inhabiting the human oral cavity, including pathogenic strains, has been previously assessed. The inhibitory activity of chitosan (MIC = 0.16 mg/mL) and the *Scutellaria baicalensis* extract (MIC > 2.50 mg/mL) was demonstrated against the growth of *Streptococcus mutans* [26], which is indicated as the main cause of periodontitis. Additionally, the activity of the obtained scaffolds (MIC = 1.23 mg/mL) against *S. mutans* was evaluated. When compared to the extract, the results demonstrated a considerable increase in the sensitivity of the pathogens to scaffolds.

Biocompatibility is unquestionably regarded as one of the most crucial factors in the effective manufacture of scaffolds. One of the greatest ways to create more tissue-friendly hydrogel composites is to choose biocompatible components (such polymers with intrinsic antibacterial capabilities) and add non-toxic additives to the hydrogels. As a result, the suggested scaffold’s composition satisfies the requirements. Therefore, cytotoxicity was assessed using the MTT test, which was found to be reliable in assessing the biocompatibility of dental materials [42]. The 1% hydrogel concentration used in the culture medium and the appropriate concentrations of individual scaffold components did not affect the viability of fibroblasts at all (Figure 6). The obtained results confirm the known property of chitosan as a biocompatible material [43,44]. Due to the lack of influence of the 2.5CS/2Gel/10Ex scaffold on the viability of the fibroblast cell, this indicates the biocompatibility of the produced material.

Fibroblasts are essential for the healing of periodontal wounds. These cell populations are necessary for the periodontal ligament, gingiva, and tooth root to regenerate a strong fibrillar connection [45]. So, using the same concentration as in the case of cytotoxicity tests, i.e., 1% hydrogels in the culture medium and appropriate concentrations of scaffold components, the impact of the scaffold and individual components on the wound closure properties in the scratch test was assessed. After 24 h of exposure to the 2.5CS/2Gel/10Ex scaffold, fibroblasts moved toward the opening to close the scratch wound by 97.1% and significantly accelerated the wound healing process compared to that in the control (68.1%) and scaffold individual components (Figure 7 and Figure 8). Importantly, the extract itself shows a positive trend towards wound healing, closing the wound by 79.7% within 24 h. While the results for the chitosan base alone were not spectacular, previous research results indicated a significant impact of the molar mass and degree of chitosan deacetylation on the wound healing process [44]. Therefore, this justifies the choice of high molar-weight chitosan as the scaffold base in the presented study. Importantly, based on the literature data, wound healing properties demonstrated in vitro show a high correlation with in vivo studies [45].

Based on the presented results of preliminary tests, as well as the characteristics of the materials, the best printability, producing smooth and uniform scaffolds, was obtained using the hydrogel with 2.5% *w*/*v* of high molecular weight chitosan (CS), 2% *w*/*v* gelatin (Gel), and 10% *w*/*v* of extract (Ex). The results, together with the previously demonstrated properties of the combination of chitosan and the extract, render the 3D-printed CS/Gel/extract scaffolds as potential candidates for the treatment of periodontal diseases. To commercially introduce the proposed combination to the pharmaceutical market, it will be necessary to prove effectiveness in vivo, as well as biodegradability tests in the oral environment.

## 4. Conclusions

The obtained 3D dressings containing CS/Gel/extract systems meet the criteria for personalized treatment of periodontal diseases due to their functionality (proven anti-inflammatory effect) and form (preserving shape with long-term release of active compounds). The amorphous state of the 3D dressings containing CS/Gel/extract systems was established by XRPD measurements, while the uniform distribution of hydrogel within the scaffold was proved by microscopic analysis. Furthermore, FTIR-ATR measurements defined interactions between the chitosan-based scaffold and the active compounds of the Baikal skullcap root extract. Finally, their anti-inflammatory, microbiological, and wound healing activities, together with biocompatibility properties, ensure safety and biological effectiveness, supporting their potential in tissue engineering applications.

## Figures and Tables

**Figure 1 pharmaceutics-16-00359-f001:**
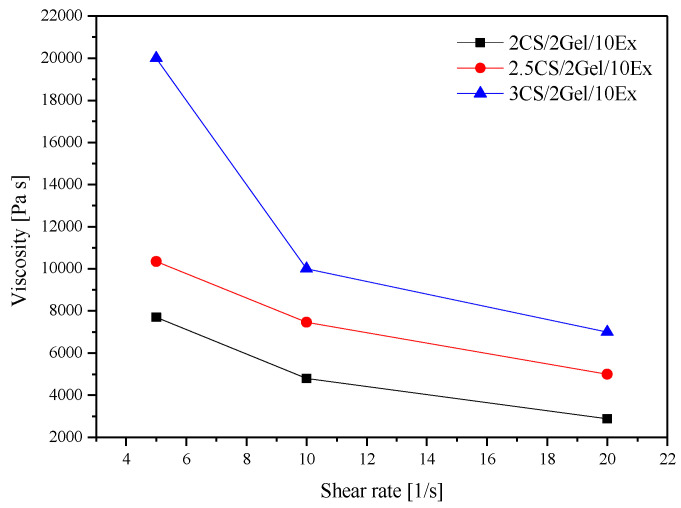
Hydrogel viscosity at 25 °C.

**Figure 2 pharmaceutics-16-00359-f002:**
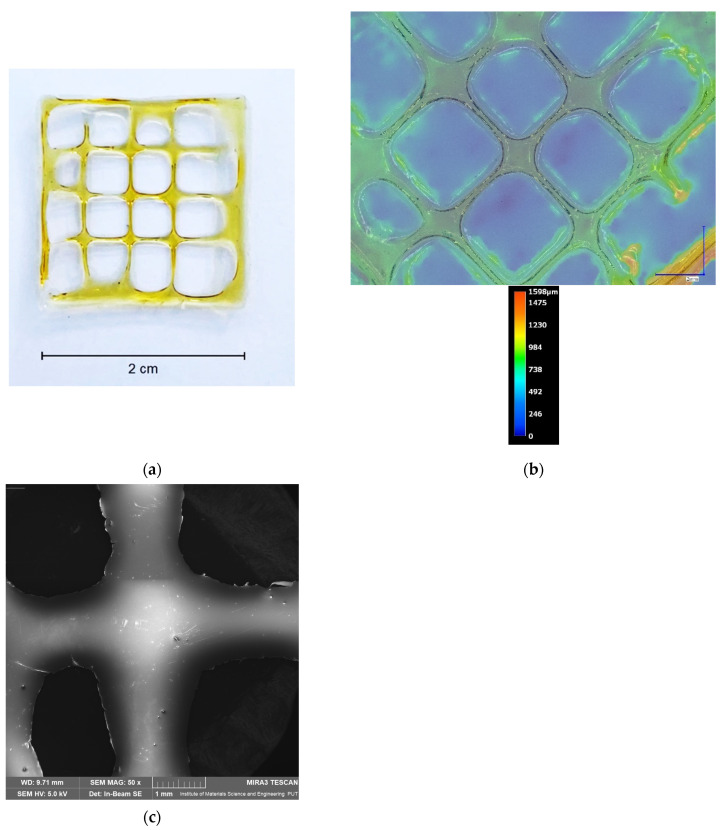
Photos of dried 3D-printed multilayered 2.5CS/2Gel/10Ex scaffold (**a**), its microscopic (**b**), and SEM (**c**) images.

**Figure 3 pharmaceutics-16-00359-f003:**
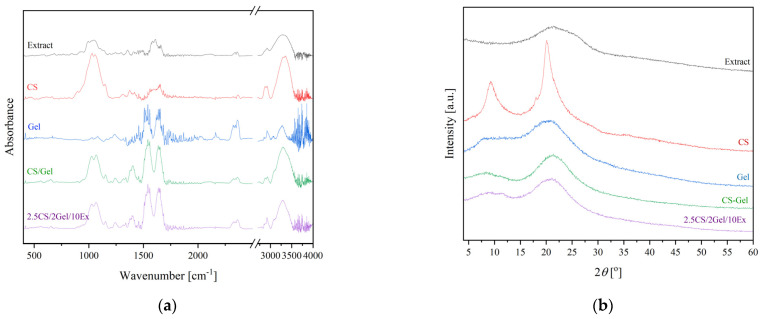
ATR-FTIR spectra (**a**) and XRPD spectra (**b**) of extract (black line), CS (red line), Gel (blue line), 3D-printed base (green line), and 3D-printed 2.5CS/2Gel/10Ex scaffold (purple line).

**Figure 4 pharmaceutics-16-00359-f004:**
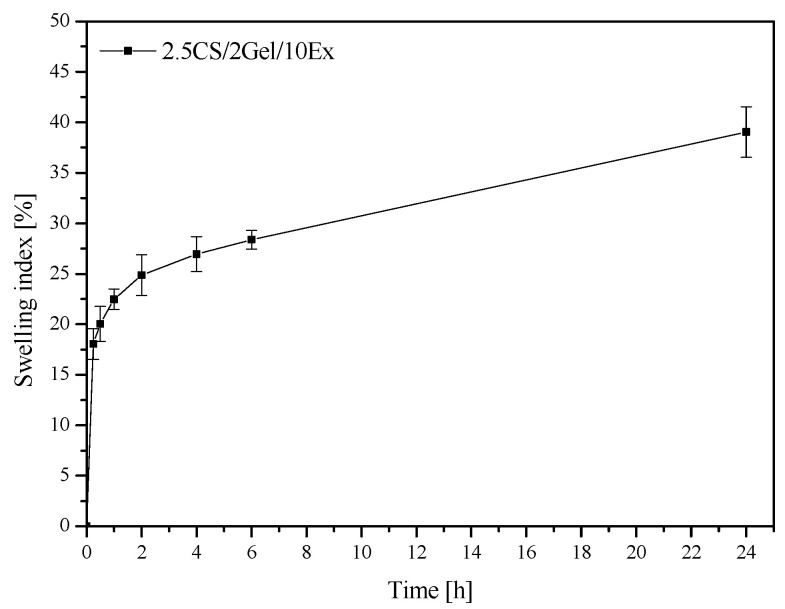
Swelling behavior of 2.5CS/2Gel/10Ex scaffold.

**Figure 5 pharmaceutics-16-00359-f005:**
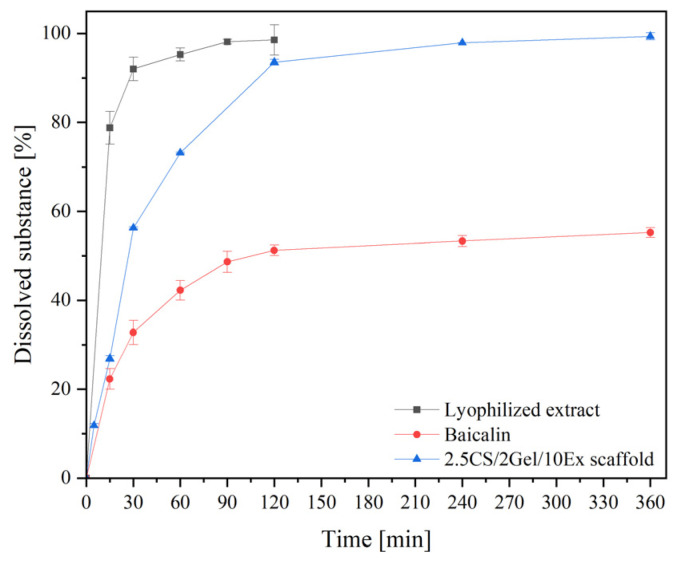
In vitro release profile of Baicalin from 2.5CS/2Gel/10Ex scaffold.

**Figure 6 pharmaceutics-16-00359-f006:**
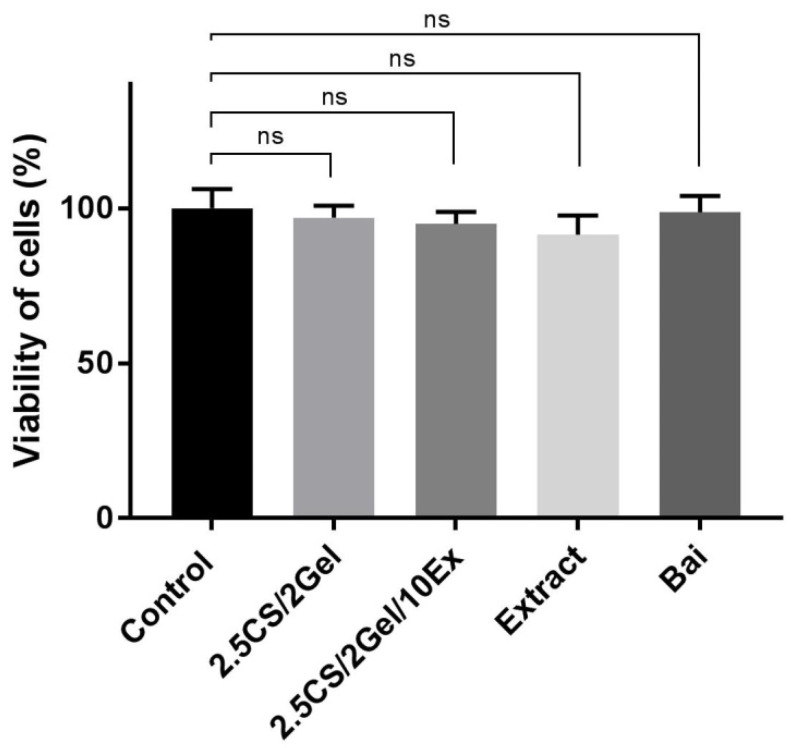
Viability of human normal skin fibroblasts (Hs27 cells) incubated for 24 h with hydrogel base (2.5CS/2Gel), hydrogel-based scaffold (2.5CS/2Gel/10Ex), *Scutellaria baicalensis* extract (Extract), and baicalin (Bai). The samples were examined by MTT test at the concentrations used in the wound healing assay. Results were statistically analyzed by ANOVA with a post hoc Tukey’s test. ns—not significant (*p* > 0.05).

**Figure 7 pharmaceutics-16-00359-f007:**
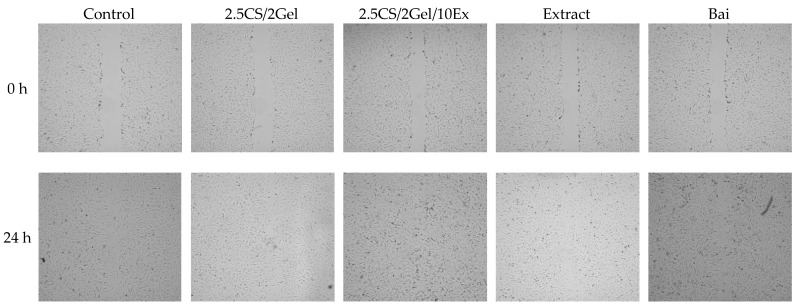
Representative images of wound closure effect of hydrogel base (2.5CS/2Gel), hydrogel-based scaffold (2.5CS/2Gel/10Ex), *Scutellaria baicalensis* extract (Extract), and baicalin (Bai).

**Figure 8 pharmaceutics-16-00359-f008:**
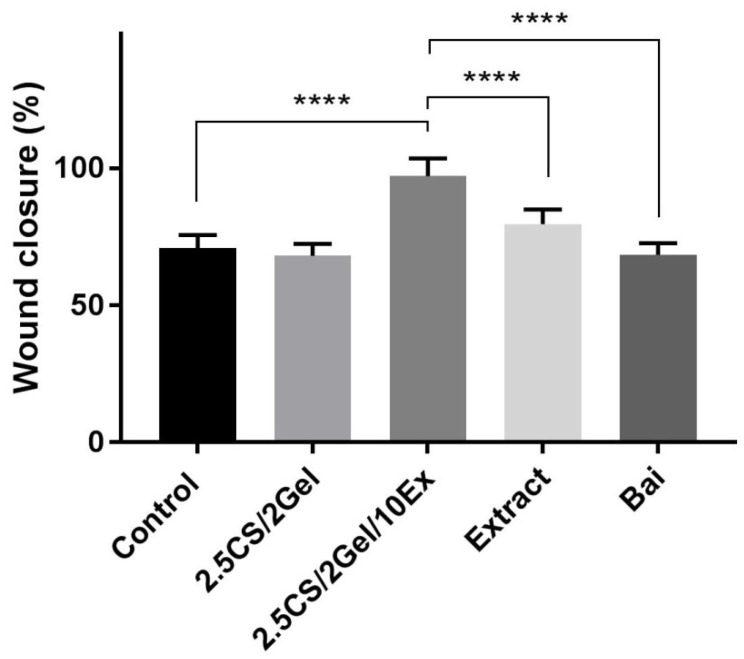
Wound healing activity observed after 24 h incubation of human normal skin fibroblasts (Hs27 cells) with hydrogel base (2.5CS/2Gel), hydrogel-based scaffold (2.5CS/2Gel/10Ex), *Scutellaria baicalensis* extract (Extract), and baicalin (Bai). Samples of 2.5CS/2Gel and 2.5CS/2Gel/10Ex were mixed with the culture medium to obtain 1% solution (0.1 mL of 2.5CS/2Gel or 2.5CS/2Gel/10Ex + 9.9 mL of medium). Concentration of Extract was adjusted to its content in 2.5CS/2Gel/10Ex, while Bai was tested at its concentration determined in *Scutellaria baicalensis* extract. Results were statistically analyzed by a one-way ANOVA with a post hoc Tukey’s test. Statistical significance was designated as **** when *p* < 0.0001.

**Table 1 pharmaceutics-16-00359-t001:** Composition of 3D-printed scaffolds.

	2CS/2Gel/10Ex (P1)	2.5CS/2Gel/10Ex (P2)	3CS/2Gel/10Ex (P3)
Extract	10% *w*/*w* *	10% *w*/*w* *	10% *w*/*w* *
CS	2% *w*/*v*	2.5% *w*/*v*	3% *w*/*v*
Gel	2% *w*/*v*	2% *w*/*v*	2% *w*/*v*

CS—chitosan; Gel—gelatin; * in relation to CS content.

**Table 2 pharmaceutics-16-00359-t002:** The required pressure of the prepared hydrogels.

	2CS/2Gel/10Ex (P1)	2.5CS/2Gel/10Ex (P2)	3CS/2Gel/10Ex (P3)
Minimum required pressure [kPa]	44 ± 2	85 ± 2	135 ± 3

**Table 3 pharmaceutics-16-00359-t003:** Average scaffold’s uniformity factor and pore factor.

	Uniformity Factor, *U*	Pore Factor, *Pr*
2CS/2Gel/10Ex (P1)	1.4852 ± 0.0147 ^b^	1.1981 ± 0.0485 ^a^
2.5CS/2Gel/10Ex (P2)	1.0415 ± 0.0624 ^a^	1.0456 ± 0.0681 ^a^
3CS/2Gel/10Ex (P3)	1.0748 ± 0.0268 ^a^	1.1095 ± 0.1162 ^a^

Using Duncan’s test, mean values within a column with the same letter do not differ substantially at *p* < 0.05; the lowest values are represented by the first letter of the alphabet, and statistically significant falling values by the following letter.

**Table 4 pharmaceutics-16-00359-t004:** Parameters of mathematical models fitted to the baicalin release profiles from 2.5CS/2Gel/10Ex scaffold.

	Zero-Order Kinetic		First-Order Kinetic		Higuchi Kinetic		Korsmeyer–Peppas Kinetic	
	K	R^2^	K	R^2^	K	R^2^	n	R^2^
2.5CS/2Gel/10Ex	15.82	0.68	0.48	0.40	**15.01**	**0.95**	0.62	0.81

The highest degree of correlation is marked in bold.

**Table 5 pharmaceutics-16-00359-t005:** Anti-inflammatory activity of CS, extract, and 2.5CS/2Gel/10Ex.

	Hyaluronidase Activity Inhibition IC_50_ [mg/mL]
CS	0.50 ± 0.01
Extract	12.96 ± 0.56 [22]
2.5CS/2Gel/10Ex	63.57 ± 4.98

## Data Availability

All data supporting the reported results can be found within the manuscript.

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
