# Peer review of "3D-Printed Chitosan-Based Scaffolds with *Scutellariae baicalensis* Extract for Dental Applications"

_pharmaceutics, 2024, doi:10.3390/pharmaceutics16030359_

Round 1

Reviewer 1 Report

Comments and Suggestions for Authors

Dear Authors, 

The 3D dressings made with CS/Gel/extract systems have been shown to have the necessary qualities for the personalized treatment of periodontal diseases. This is because they exhibit both functional (demonstrated anti-inflammatory properties) and structural (maintaining shape while releasing active compounds over a long period) characteristics. XRPD measurements confirmed that these dressings are in an amorphous state. Additionally, FTIR-ATR revealed interactions between the chitosan-based scaffold and the Baicalin. Lastly, their anti-inflammatory properties demonstrate biological effectiveness, further highlighting their potential for use in tissue engineering applications.

Consideration must be given to addressing the following points:

·       Line 74; It should be clarified that the term "ECM" means "Extracellular medium".

•  Line 104: By stating “1% in 1% v/v acetic acid” it seems that chitosan was provided as a solution.

•  Line 107; please specify the components of the “artificial saliva solution”.

•  Anti-inflammatory activity method should be explained.

•  Line 174; Please indicate the HPLC condition method.

•  Since thickness is an important attribute of the scaffold, it is essential to report the thickness of the “3D-printed multilayered 2.5CS/2Gel/10Ex scaffold”.

•  Please describe the “Anti-inflammatory activity test” in the method section.

•  Line 349; It appears that the uniformity of the active compound cannot be verified by microscopic analysis. It is recommended to review this section.

•  In this study, the equal-sized square-shaped mesh scaffold and the uniformity of the thickness of the produced dressing are important characteristics, so it is necessary to report the uniformity factors during 360 h of the In vitro release profile of Baicalin.

Regards, 

Author Response

Please see the response in the attachment

Reviewer 2 Report

Comments and Suggestions for Authors

The manuscript entitled “3D-printed chitosan-based scaffolds with Scutellariae baicalensis extract for dental applications” developed a chitosan-based hydrogel with Scutellariae baicalensis extract, compatible with 3D printing. However, since the same group has published in Pharmaceutics 2022 demonstrating chitosan as a functional carrier for the local delivery of Scutellariae baicalensis radix extract, this current paper should focus more on the 3D printing aspect for possible publication.

Comments:

1.     Table 1: The authors only used one weight ratio of the extract through all of the groups. Please explain the rationale of using 10% w/w extract in the hydrogel. Would this weight ratio lead to obvious cytotoxicity or a higher weight ratio of the extract can be achieved? Furthermore, the variation in chitosan or gelatin is also minor compared to other studies. Please provide more discussions/clarification in how to determine the composition of scaffolds. (major)

2.     Figure 2:

1)     It seems that the authors did not clarify how many layers were printed to fabricate a single scaffold. If more than one layer were printed, then what’s the maximal layers/height that can be achieved with this composition? (Major)

2)     Based on figure 2 (a), the reviewer assumes that the authors aimed at printing scaffolds with a filament gap around 500 microns. Is it practical to print a narrower filament gap, such as 250 microns or smaller? (Minor)

3.     While the title is targeting dental/periodontal applications, there are zero in vitro/in vivo experiments conducted in dental/periodontal cell lines or related animal models. Please refer to Yao, Y., et al. Journal of Dental Research 101.12 (2022): 1457-1466. for more guidance in proliferation/differentiation/cytotoxicity/coculture experiments targeting periodontal-related cell lines. (major)

4.     Language should be thoroughly revised. For example, during the abstract - “a significant Increase In the release of baicalin in vitro was observed” was stated. The authors are supposed to be more careful in differentiating letter cases.

Comments on the Quality of English Language

The quality of English Language should be improved.

Author Response

(The authors gave the same response as above.)

Reviewer 3 Report

Comments and Suggestions for Authors

In this study, authors developed a 3D-printed chitosan-based hydrogel rich in Scutellariae baicalensis extract as an innovative approach for the personalized treatment of periodontal diseases. Authors stated that the results render the 3D-printed CS/Gel/extract scaffolds as potential candidates for the treatment of periodontal diseases. The following issues should be addressed.

1. Please provide more characterization results of the 3D-printed CS/Gel/extract scaffolds, like SEM to show the surface morphology.

2. How to prove the periodontal diseases treatment potentials of the scaffolds? It seems that there was no periodontal diseases results in the paper.

3. How about the biocompatibility of the 3D-printed CS/Gel/extract scaffold? This should be studied in the paper.

4. Related papers on hydrogels can be cited: Mussel-inspired “all-in-one” sodium alginate/carboxymethyl chitosan hydrogel patch promotes healing of infected wound; Radiation synthesis and characterization of nanosilver/gelatin/carboxymethyl chitosan hydrogel

Author Response

(The authors gave the same response as above.)

Reviewer 4 Report

Comments and Suggestions for Authors

The article was interesting because the problem of periodontal changes is a very big challenge for dentistry, which is why any work in this field is highly desirable.

  Fro my side, congratulations on a well-written and interesting work, but I have a few comments that require clarification:

Introduction

It would be good to add 1-2 more sentences about how periodontal lesions are currently treated to show how innovative your work is.

Secondly, before any material is placed into the gingival pocket, it must be cleaned and eliminated of bacteria, otherwise the treatment will not be effective.

Line 69

Furthermore, CS promotes hemostasis due to direct electrostatic interactions between negatively charged red blood cells and platelets and the positively charged molecules of CS, leading to blood coagulation- instead of using the abbreviation CS twice in one sentence, could you use chitosan, a polysaccharide?

Line 74

ECM- If you are using an abbreviation for the first time, it is good to explain what it means, thank you

As you know, plant raw materials often have a very diverse composition, which varies depending on the year and growing conditions, so in your further work you should consider appropriate analysis of this raw material in terms of uniform properties.

At the end of the introduction, you can also put forward a thesis about what to expect from conducting your research?

Materials and methods

Line 100

Has the extract been tested in any way for what it contains?

were stirred for 15 minutes at 4000 rpm to remove any air bubbles.- centrifuge? Producer country?

Line 118

extract was added into a gelatin aqueous solution under continuous magnetic stirring- how long ? at room temperature?

If you want to print this scaffold, it must be designed in a computer (shape, size, thickness, etc.) In what program?

Line 145

VHX 7000 Keyence- country?

Line 153

. X-ray Powder Diffraction-and what powder did you measure using diffraction? Did you expect there to be any crystal structures?

Line 168

Agilent 708- DS dissolving equipment.- producer country?

Line 181

Statistica 13.3- producer country?

Results and discussion

Line 202

Therefore, initially, the 3 new-formed formulations were tested for their viscosity

through rheological measurements via increasing the shear rate (Figure 1- but in the materials and methods section you didn't mention anything about measurements using a rheometer? What type of boiling point, speed etc, you have to add it to the M&M part, please.

Line 256

Characteristic bands of extraction and CS have been previously described- characteristic peaks in the IR spectrum are known for CS and the extract tS. baicalensis

Figure 3

Blue line Gel it means gelatin?

What are the limitations of your research? What else would have to be done in the future so that such support could be implemented into the government?

For the discussion you can you following ref:

https://pubs.rsc.org/en/content/articlehtml/2023/bm/d3bm00719g

good luck in further research!

Author Response

(The authors gave the same response as above.)

Reviewer 5 Report

Comments and Suggestions for Authors

This paper aims to develop a 3D-printed chitosan-based hydrogel rich in Scutellariae baicalensis extract as an innovative approach for the personalized treatment of periodontal diseases. It was found that render the 3D-printed CS/Gel/extract scaffolds as potential candidates for the treatment of periodontal diseases, through the characterization of some properties of 3D-Printed Scaffolds.

It is a well-structured and organized document.

Abstract and material and methods should include the sample size or how many independent experiments were performed in each assay

Introduction – please develop some crucial areas of this introduction like the clinical use of this kind of treatment of periodontal disease; in what phase of the periodontal treatment is to be intended to be use, in conjunction or solely to other treatments. Also the exact location of this scaffolds should be explore with examples of other compounds used and respective results. Why is important to create this innovative treatment. Clinical Indications of this kind of treatment also could help clinical judgement and importance of this treatment.

Material and Methods- should include the sample size or how many independent experiments were performed in each assay.

Explain why it was not used periodontal fluid instead of articial saliva, since has different chemical composition and pH values.since it is applied in endo osseous area, why is leacheble in saliva articial instead of blood.

Subtitle 2.5.5. expected active componennts shoul be indicated

Results and discussion- limitations of the study should be adress, as also the indication of future prospectives like stability of the scaffolds to the biodegradation processes that exists in the oral cavity. Have difficult realizing haw this treatment can be personalized to each patient oral cavity.

Author Response

(The authors gave the same response as above.)

Round 2

Reviewer 2 Report

Comments and Suggestions for Authors

The reviewer thanks the authors' efforts in addressing all of the questions raised. The manuscript is now qualified for possible publication in Pharmaceutics

Comments on the Quality of English Language

Minor improvements are needed.

Reviewer 3 Report

Comments and Suggestions for Authors

No further comments.